# Maternal Communication with Preschool Children about Morality: A Coding Scheme for a Book-Sharing Task

**DOI:** 10.3390/ijerph191811561

**Published:** 2022-09-14

**Authors:** Jéssica Rodrigues Gomes, Suélen Henriques Da Cruz, Andreas Bauer, Adriane Xavier Arteche, Joseph Murray

**Affiliations:** 1Postgraduate Programme in Epidemiology, Federal University of Pelotas, Rua Marechal Deodoro, 1160, 3 Piso, Pelotas 96020-220, RS, Brazil; 2Human Development and Violence Research Centre (DOVE), Federal University of Pelotas, Pelotas 96020-220, RS, Brazil; 3Postgraduate Programme in Psychology, Pontifical Catholic University of Rio Grande do Sul, Avenida Ipiranga 6681, Prédio 11, Porto Alegre 90619-900, RS, Brazil

**Keywords:** moral development, moral communication, children, book-sharing, mother–child interaction

## Abstract

**Background:** Preventing interpersonal violence requires understanding the moral development and determinants of child aggression. Communication about moral values and concerns by parents is theoretically important in this process. We aimed to develop a coding system to measure mothers’ communication about morality with young children and test its psychometric properties. **Method:** The cross-sectional study included a subsample (*n* = 200) of mothers and their four-year-old children in a population-based Brazilian birth cohort. Mothers and children were filmed while looking at a picture book together, containing events of aggression, taking away without asking, and several prosocial behaviours. Films were transcribed and a coding system, including 17 items, was developed to measure the maternal moral judgements and the explanations communicated to their children. Inter-rater reliability was estimated, and exploratory factor analysis performed. **Results:** Mothers judged acts of physical aggression as wrong more frequently than taking away material goods without asking; most mothers communicated about the emotional consequences of wrong behaviour with their child. Two latent factors of moral communication were identified, *interpersonal moral concern* and the expression of *material moral concern*. There was excellent inter-rater reliability between the two coders. **Conclusions:** Parent–child book-sharing provides a means to measure maternal communication about morality with their children. The coding system of this study measures both communication about *interpersonal moral concern* and *material moral concern*. Further studies with larger samples are suggested to investigate the importance of these dimensions of caregiver moral communication for children’s moral development.

## 1. Introduction

Interpersonal violence is a major global health problem with a range of environmental determinants starting early in life. A major area of research has been on determinants of child aggressive behaviour, which is a strong predictor of violence and that can be understood in a more general context of child moral development. Parental communication about moral values and concerns is theoretically important in this process. According to an important framework proposed by Tomasello [1], moral development involves constructing an understanding of why different actions are right or wrong and behaviours that encompass respect for others, concern for the wellbeing of others, care, cooperation, respect for other people’s rights, and notions of fairness [2]. Although the role of parenting has been amply studied in relation to children’s moral development, this has mostly focused on the influence of parental behaviours in terms of discipline practices, sensitivity, and care. However, less is known about the role of parental communications in conversations about moral issues, and few measures [3,4] are available to assess parental moral communications to further this line of research.

The role of caregiver behaviour and communication is a common focus for several key theories of moral development. For example, according to Bandura’s classic social learning theory [5], parents teach their children moral standards by guiding their behaviour and explaining the patterns of conduct that are considered appropriate. As such, parents’ values and moral understanding are shared with children, and children use them as a basis for their own moral judgments [3]. Several theories have proposed that children’s moral development occurs in stages and proposed determinants of this evolution over time [6,7,8,9]. Early childhood is considered a critical period for moral development [10] in which the family is the primary socialising environment and in which children internalise various learnings about the social world [11] and rapidly develop a core understanding of interpersonal relationships and behaviours [9,12,13]. According to Vygostsky [14], as children develop language skills and engage in conversations with their parents, they learn to represent their experiences in an organised way, thus developing a core understanding of the social environment, including its moral aspects [13]. Turiel’s [15] classic social domain theory [16] divides children’s social development into several domains—moral, conventional, and personal—with moral development influenced by concerns around well-being and care for others, particularly influenced by parental behaviour and communication early in life.

Recent empirical research has highlighted the potential role of communication between parents and children for moral development [9,10,17,18,19,20]. Partly through conversational interactions with parents, children start to develop a basis for interpreting what is acceptable or unacceptable behaviour [21]. Both directly and indirectly, parents teach their children rules: for example, they may help children understand the consequences of their or others’ actions and encourage them to do what they think is right, while rebuking mistakes and praising acts of kindness [9]. Such conversations provide a space for children to expand and alter their understandings, with the integration of new ideas and views and provides a context for children to develop their moral understanding [9,22]. Particularly from about age three years, verbal development means that parents’ communication about social interactions and moral issues plays a fundamental role in moral development [1]. Corroborating the importance of parental moral values imported to children, a study in the United States found that parental values regarding fairness and wellbeing were associated with young children’s moral preferences as well as neural differences while perceiving helpful versus harmful social behaviours [23].

In a Canadian study, 100 pairs of mothers and children (aged 7 to 11) were filmed while discussing past events in which the child helped a friend and he/she hurt a friend [4]. The study showed that conversations about help facilitated children’s perceptions of themselves in prosocial ways and encouraged them to practise helping behaviour. The study suggested that when mothers aided their children to recognise that their actions had resulted in harmful emotional consequences for others, this promoted children’s understanding of how to recognise the others’ needs and taught them sensitivity about harm to others. Similarly, in a longitudinal study in the United States, conversations between mothers and their 3–5-year-old children about past events was associated with children’s improved emotional understanding by providing linking emotions with experiences of the child and those of others [24].

Furthermore, a cross-sectional study in the United States of four-year-old children found that those whose mothers discussed people’s feelings more frequently and used evaluative terms such as “good boy” and “this is the right thing to do” were more advanced in terms of development of conscience [25]. Likewise, a longitudinal study in the United States evaluated the communications of 66 mothers and their 2½-year-old children during a conflict situation and in conversation about the child’s past behaviour. Situations in which the mothers explained the conflict situation and mentioned resolution strategies were predictive of socioemotional and moral development among their children six months later [26].

Studies have also found that specific aspects of parental communication are important for different types of moral behaviour. This has been examined in many studies using book-sharing tasks to observe communication between parents and children—a method considered to effectively stimulate communication and provide hypothetical models of moral behaviour of various types [27,28]. In such studies, some forms of communication (using explanations involving emotions) specifically correlated with prosocial behaviour [17,18,19,29,30,31], whereas others (confused explanations about moral situations) were more associated with child aggression [3,30]. However, the evidence is still scarce and mainly comes from small studies and exclusively from high-income countries. There are no known Brazilian studies on this subject.

A difficulty in this area of research is that there are many different coding schemes for evaluating the content of parental communication about morality that have been used between individual studies without adequate psychometric development. Most coding schemes were created in line with common communications found in the individual study sample, and in light of the specific objective of each study, leading to coding systems with diverse characteristics. Ideally, it would be explored how different aspects of parental moral communication inter-relate and their factor structure—whether there is just one general domain of moral communication or different constructs reflecting specific aspects of moral judgement and concerns—relate to fairness and others to interpersonal well-being.

One coding scheme used across several studies was created by Recchia et al. (2014) [4] with the following codes for mothers’ communication with their children about experiences of helping and aggression: moral judgment of behaviour (positive or negative), reasons for the behaviour (including motives and emotions), consequences of the behaviour (punitive, physical, emotional, and relational) and strategies to manage similar situations (such as making reparations or asking for intervention from an adult). Similarly, in evaluating parents’ communication during book sharing, Brownell et al. (2012) [18] coded content relating to emotions, explanations of emotions, mental states, and statements that promoted empathy with the actors’ emotions. However, neither study examined the question of whether these different communications form singular or multiple, different domains.

It should be emphasised that different socioeconomic and cultural contexts are likely to influence parental communication with their children about moral norms and values [32], and consequently specific items that may be needed in a coding scheme about parental moral communication. The current study was conducted in Brazil where we are not aware of any prior instrument having been applied to evaluate the communication of parents with their children about moral issues. Thus, it was considered important to construct and test a coding scheme for use in the Brazilian population based on theories on moral development and considering prior relevant coding schemes [4,7,33] and any new types of moral communications observed in a Brazilian sample. To this end, the current study investigated maternal communication about moral issues in a subsample of the 2015 Pelotas Birth Cohort in which mothers and their four-year-old children were filmed while performing a book-sharing task.

## 2. Methods

A cross-sectional study was conducted nested in the 2015 Birth Cohort in Pelotas, Rio Grande do Sul, Brazil. This cohort includes 4275 children who were born alive in the urban area of Pelotas in 2015 (99% of all lives births in the city that year) and their mothers, and various health and psychosocial aspects of development through the first years of life were evaluated. To date, the cohort has had six assessment waves: prenatal in 2014/2015, perinatal in 2015, at three months old in 2015/2016, twelve months in 2016, twenty-four months in 2017, and four years in 2019. The methodologies used are detailed in other publications [34].

The present study used data from the sixth follow-up of the cohort when the children were four years old. At this stage, all the children participating in the cohort, as well as their mothers, were invited to participate. The response rate of this follow-up was 95.4% (*n* = 4010). Data collection occurred at the Epidemiological Research Centre of the Federal University of Pelotas. Questionnaires were applied by trained and qualified interviewers who also filmed several parent–child interaction tasks and made other measurements. The questionnaires were administered using the Redcap instrument [35]. Physical health exposures and outcomes were evaluated along with a wide range of assessments of parenting, child psychosocial development, and family and social environments. The main data analysed in the present study came from the filmed book-sharing task, which occurred after a period of free play early in the assessment.

### 2.1. Ethical Considerations

Before each assessment wave, participants signed an informed consent statement containing full explanations about the research. Anonymity and data confidentiality were guaranteed, and participants were informed that they had the right to drop out from the study or refuse to participate at any stage. The evaluations of the 2015 Pelotas Birth Cohort between zero and four years of age were approved by the Research Ethics Committees of UFPel (School of Physical Education protocol #26746414.5.0000.5313; and Faculty of Medicine protocol #03837318.6.0000.5317).

### 2.2. SubSample Used in the Present Study

A subsample of 200 mother–child pairs were selected for the current study. This subsample was selected by first stratifying the whole cohort with filmed book-sharing video data (*n* = 3865) into five socioeconomic quintiles and then randomly selecting 40 mother–child pairs from each stratum. Encoding a random proportion of data has the aim of obtaining a sample that represents the larger population [36].

### 2.3. Filmed Book-Sharing Task: Data Collection

In the filmed book-sharing task, a book called “A Day at the Park” was used which contains only pictures, without text, and was specially developed by the research team with the aim of representing children’s social activities and eliciting conversations about moral issues pertinent to young children. The interviewer gave the book to the mother and asked her to look at with the child in a natural manner, just as she might do at home. Mothers and children were filmed for approximately five minutes during this activity without any interference from the examiners. Afterwards, the dialogue of the footage was transcribed for applying the coding system developed in the present study.

The book presents images of three actions of aggression and taking away without asking. Action 1 involves a child pushing another child; action 2 involves the same child taking the other child’s toy without asking; and in action 3, the first child’s mother attempts to apply physical punishment to the child who had taken the toy. In addition, several actions exemplifying prosocial behaviour are shown, for example a child asks for help from her mother, another shows children sharing their toys, and another shows a mother talking with a child to resolve the conflict. Four sample images from the book are shown in Figure 1, and a copy of the whole book is available in PDF format on request from the corresponding author of this article.

### 2.4. Filmed Book-Sharing Task: Transcription and Development of the Coding Scheme

The book-sharing videos were transcribed into Excel with each verbal phrase entered on a new line by trained transcribers. For quality control, monthly meetings were held in which all the transcribers worked on the same video, and any inconsistencies were discussed.

The coding system for maternal moral communication was developed using thematic analysis, as proposed by Braun and Clarke (2006) [37]. For effective development of the system, a deductive approach was used which prioritised a solid theoretical foundation before data verification [36]. Accordingly, first, psychological theories that already exist are identified and condensed into more pertinent themes that help direct identifying themes within the data collected in a new study. For the current study, first, themes about moral communication were identified from the literature (as reviewed in the Introduction) to develop a provisional coding plan. All codes from prior studies were considered, but the main scheme that the current study was based on was that of Recchia et al. (2014) [4] with codes for: moral judgment of behaviour (positive or negative), reasons for the behaviour (including motives and emotions), consequences of the behaviour (punitive, physical, emotional, and relational), and strategies to manage difficult behaviours (such as making reparations or asking for intervention from an adult). After building an initial list of possible codes based on the literature, 200 transcripts were then read from the current study [37] and redundant codes (communications never mentioned) were eliminated, and any new codes (communications about moral issues observed in the current study but not captured by previous coding systems) were identified and added to the current coding scheme. As such, a complete list of codes was created for the current study, and example transcript phrases for each code were recorded. After developing this initial coding system, a testing phase was conducted using 10 transcripts. Two postgraduate students independently coded 10 transcripts. Subsequently, they discussed the results with regard to any divergences in the specific codes assigned until they reached a consensus [38]. It was not found necessary to make any further adjustments to the coding scheme.

### 2.5. Final Coding System

The final coding system applied in the current study to the subsample of 200 mother-child pairs included categories of maternal moral communication for each of the aggressive and prosocial actions shown in the book, receiving the code “0” when certain content was not present in the mother’s speech and “1” when this content was present. For the three actions of aggression, the following contents were coded: (a) whether the mother judged the aggressive behaviour as wrong; (b) whether the mother mentioned reasons why the aggressive behaviour occurred; (c) whether the mother attributed punitive consequences to the aggressive behaviour; (d) whether the mother attributed physical/material consequences to the aggressive behaviour; and (e) whether the mother attributed emotional consequences to the aggressive behaviour. In addition, the coding system covered maternal communications relating to prosocial behaviour shown in the book (actions 4): (f) whether she spoke in a way that encouraged conflict resolution strategies based on conversation; and (g) whether she encouraged sharing or helping behaviour. Thus, the coding system had a total of 17 items. Table 1 presents the encoding system with examples for each item. To test the reliability of the data, the two postgraduate psychology students coded all 200 transcripts.

### 2.6. Data Analysis

First, descriptive statistics of the sample and items from the filmed book-sharing task were calculated. Second, the level of agreement between the coders was then analysed by calculating kappa for each item in the moral-communication-coding scheme. The results (size of kappa) were interpreted as follows: <0.00 = poor; 0.00 to 0.20 = weak; 0.21 to 0.40 = acceptable; 0.41 to 0.60 = fair; 0.61 to 0.80 = good; 0.81 to 0.99 = excellent; and 1.00 = perfect [39].

Third, an exploratory factor analysis was conducted on the 17 items coded in the book-sharing task using the statistical program, Mplus, version 8.7, Muthén Muthén, Los Angeles, CA, USA [40]. This is a method widely used in research in the field of psychology [41] to determine the nature and number of latent factors that best represent the dataset [42]. To identify the best-fitting model for the dataset in the current study, one to four factors were tested, using oblique geomin rotation, as estimated through WLSMV [43]. It is considered that an acceptable fit has RMSEA between 0.05 and 0.08; SRMR between 0.05 and 0.10; CFI greater than or equal to 0.95; and TLI greater than 0.9 [43,44]. To determine the ideal number of factors, eigenvalues greater than or equal to one were considered [45]. Factor loadings of individual items were considered acceptable and included in the final model when they were above 0.20 [46]. When an individual item had a factor loading > 0.20 for multiple factors, it was assigned to the factor for which it had the greatest factor loading.

## 3. Results

Table 2 presents the sociodemographic characteristics of mothers in the study. Most mothers were aged between 20 and 30 years (67.0%), with white skin colour (72.0%), had a partner (75.4%), had worked outside the home since the child turned two years old (67.3%), and about half had a family income of 1.1 to 3 minimum monthly wages (50.5%). About one third of the sample (32.0%) had 12 years or more of schooling; more than two thirds of the mothers (77.9%) characterised their own health as good to excellent; and 14.6% presented a risk of moderate to severe depression.

Table 3 shows the frequency of items in the book-sharing coding system and the level of observed agreement and concordance (and kappa statistic with standard error) between the two coders. Among the 17 variables of the coding system, the observed level of agreement ranged between 96.5% and 100% and the kappa statistic ranged from 0.94 to 1.00. Thus, concordance was classified as excellent to perfect [39].

Considering the frequency of the mothers’ moral communications, more than half the mothers (62%) judged the behaviour of pushing another child (Action 1) to be wrong, but less than half (45.5%) cited reasons why the child behaved like this. Most mothers (64%) attributed emotional consequences to the aggressive behaviour, while physical and punitive consequences were mentioned less frequently (12.0% and 4.5%, respectively).

Considering the behaviour of taking the other child’s toy without asking (Action 2), only one third of the mothers (33%) communicated to their children a judgement that this act was wrong, and fewer still mentioned reasons why the child might have behaved like this (24%). Considering consequences of this action, 37.5% mentioned emotional consequences, but very few mothers talked about physical/material consequences (4%) or punitive consequences (2%).

Regarding the aggressive discipline used by one mother in the story (Action 3), about a third of mothers (30%) communicated to their children that they judged this behaviour to be wrong; almost the same proportion suggested reasons why this had happened (33%). Again, only a small proportion (6%) of mothers talked about emotional consequences of this action, and very few (1.5%) talked about punitive consequences; no mother talked about physical consequences. Regarding encouragement of prosocial behaviour (Actions of type 4), about one fifth of the mothers mentioned sharing or helping behaviour (21%) and conversation strategies, such as conflict resolution (18%).

To examine whether maternal moral communication had more than one dimension, factor analysis was conducted. The following five items with very small or zero counts or statistically indistinguishable (correlations of +1 or −1) were not included in the factor analysis: punitive consequences of pushing, punitive consequences of taking the toy without asking, punitive consequences of maternal aggression, physical consequences of taking the toy without asking, and physical consequences of maternal violence. Thus, 12 items remained for analysis. Considering statistical parameters of the model and the theoretical coherence of the items in each factor, a two-factor model was the most appropriate, and the following acceptable indices were obtained: RMSEA = 0.054; SRMR = 0.09; CFI = 0.950; and TLI = 0.924. Table 4 shows the factor loadings for each item.

Considering inter-rater reliability of each of the individual 12 items, the observed levels of agreement ranged between 96.5% and 100%, and the kappa statistic ranged from 0.94 to 1.0

There were eight items with loadings higher than 0.2 included in Factor 1. This factor was named expression of “interpersonal moral concern” and consisted of the following: judgment of pushing, physical consequences of pushing, emotional consequences of pushing, judgment of maternal violence, emotional consequences of maternal violence, reasons for maternal violence, encouragement of helping or sharing behaviour, and encouragement of talking as a conflict management. Factor 2, called expression of “material moral concern”, consisted of four items: reasons for pushing, judgment of taking the toy without asking, reasons for taking the toy, and emotional consequences of taking the toy without asking. Figure 2 shows the correlation matrix between the items that remained in the final model.

## 4. Discussion

This study developed a new coding system for characterising mothers’ communication with preschool children about situations of aggression, taking away without asking and prosocial behaviours depicted in a picture book in an urban population in southern Brazil. There was important variation in whether mothers judged different actions as wrong, whether they identified consequences of aggressive acts, or gave explanations for such actions; and there was variation in the extent to which mothers spoke positively about prosocial behaviours depicted in the book. We identified two latent dimensions of maternal moral communication in the current study: expressions of “interpersonal moral concern” and expressions of “material moral concern”. Inter-rater reliability for coding the 12 items measuring these dimensions was very high.

The moral judgement most often communicated by mothers was that the action of one child pushing/hitting another child in the picture book was wrong. According to the social domain theory [16], situations of physical aggression represent a direct moral action in which children can easily identify negative consequences for other people’s wellbeing in terms of physical hurt. In another Brazilian study by Valadares (2019), children were presented with two (im)moral actions: one direct (hitting a friend) and the other indirect (taking a friend’s snack). Children perceived the direct moral action as more serious, justifying their judgment in relation to the wellbeing of the other person. Even by age four years, children understand situations of direct physical harm as more wrong than other situations [47]. This might make it more likely that mothers of young children focus on situations of direct physical harm in terms of moral communications. Moreover, the act of one child pushing another may represent a situation of bullying, which has growing recognition as a major problem in Brazil [48]. According to one study of children with an average age of eight years, most caregivers perceive that bullying is becoming more frequent and that physical aggression is a common problem [49]. This could explain the high frequency with which mothers expressed moral judgement about this type of behaviour shown in the book in the current study, consistent with another study in Brasil in which physical aggression among children was the main issue of concern among caregivers [50].

We found that emotional consequences of aggression and taking away without asking were the ones most cited by mothers about three situations of conflict in the current study, rather than possible punitive or physical/material consequences, which is concordant with a previous study by Recchia et al. (2014) [4]. In that study, mothers discussed emotional consequences more frequently than other types of consequences of moral transgressions when talking to their children about past situations in which they had hurt or upset a friend. It is considered that when parents help children recognise their actions can result in harmful emotional consequences for others, they promote children’s understanding and sensitivity towards others’ needs (Recchia et al., 2014) [4]. Notably, punitive consequences were the types of consequences of conflict least mentioned by mothers in the current study. By focusing more on emotional consequences of moral actions, mothers may be contributing to their children paying more attention to the wellbeing of others rather than considering moral issues primarily in relation to punitive consequences that may arise.

We found two different dimensions of moral communication in the current study, with the first characterised by *interpersonal moral concern*, composed mostly of communication about aggressive behaviours (child pushing/hitting and maternal physical discipline) and about prosocial behaviours. All items loading on this factor reflected issues relating to interpersonal relationships and the wellbeing of others. It is unsurprising that the various items concerning physical aggression were all included in a single factor, given their direct and visible consequences for the wellbeing of another person [33,51]. As mentioned earlier, these behaviours represent direct moral actions with clear harm for others, more serious than acts of taking away without asking, such as taking a toy without asking, and are considered a key issue of concern among caregivers [49]. The fact that communications encouraging prosocial behaviour also formed part of this first factor can be understood in terms of pro-social behaviour involving motives of helping other people, expressing care, or avoiding harm to others [16]. Hence, maternal communication about prosocial behaviour also expresses a core concern about interpersonal care or and others’ wellbeing. The prosocial acts in the book are also related to earlier issues of conflict in terms of representing ways to resolve conflict and avoid situations of aggression, helping understand why this range of maternal communications about aggression and prosocial behaviours formed one latent construct.

In a second dimension of maternal moral communication, a factor of *material moral concern* was identified, composed of items relating to the behaviour of taking a toy without asking. As previously mentioned, it represents a less direct type of moral harm to another person and is often considered less serious than aggressive behaviour involving direct physical harm [16,51]. Taking a toy without asking is wrong in relation to material “rights” and welfare rather than the child’s physical safety, as it is threatened by pushing/hitting or maternal physical discipline. As such, we propose that moral communication measured in this study has one dimension focussed on material moral issues and the other with the physical and emotional wellbeing of other people. It is worth noting that the item communicating “reasons for pushing” was included in the second factor of *material moral concern*, despite referring to pushing behaviour. This item had significant factor loadings in both factor 1 and factor 2, but the loading was higher in factor 2 about *material moral concern*. We believe this is because the main reason suggested by mothers for why the child pushes the other child is to obtain the other child’s toy.

The current study has several strengths. We used a stratified random sample from a population-based birth cohort to measure maternal communications about morality across different social strata. Observational data from a filmed book-sharing task were collected by trained and qualified assessors using a picture book that was specifically designed to elicit conversations about moral issues between mothers and their young children. The inter-rater reliability of the coding was excellent. The following limitations should also be considered. We were not able to run additional confirmatory factor analyses, as it was not judged appropriate to use the same participants in both models. Furthermore, as some items already had low frequencies, we decided not to split the dataset into subsamples to confirm our results using confirmatory factor analysis. The maternal communications examined in this study were based on sharing a picture book depicting a limited number of situations, and it is possible that different content would elicit additional types of communications not identified in this study.

## 5. Conclusions

To conclude, this is the first study of maternal moral communication in Brazil and a rare study of the psychometric characteristics of an instrument measuring this communication worldwide. The study found that mothers judged behaviour involving physical aggression (such as pushing/hitting and use of physical discipline) as wrong more often than behaviour relating to material goods, such as taking a toy without asking. Most mothers communicated to their children the emotional consequences of aggressive behaviour, and future research could examine the extent to which children learn from this regarding the importance of emotional wellbeing in forming moral understanding and for behavioural development. Future studies with larger samples are recommended to investigate the importance of parental communication about interpersonal moral concern and material moral concern and their influence on children’s moral development, particularly in relation to children’s aggressive behaviour, which is strongly associated with later interpersonal violence.

## Figures and Tables

**Figure 1 ijerph-19-11561-f001:**
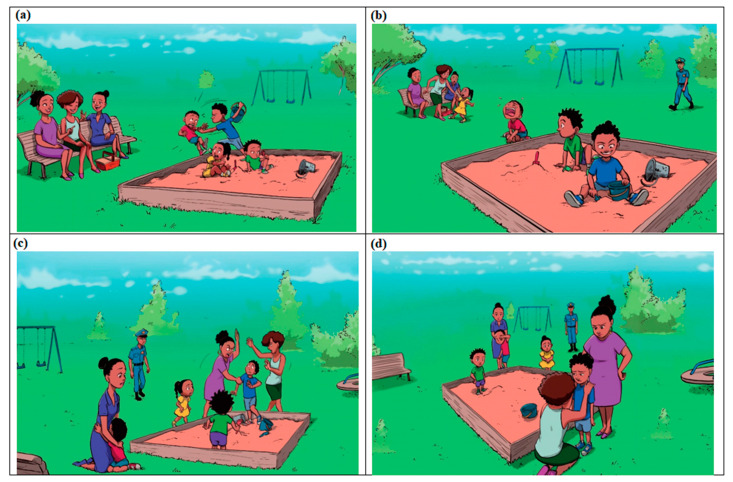
Four sample images from the book, “A Day at the Park”. (**a**) Child pushes another child [Action 1] and takes the other child’s toy without asking [Action 2]. (**b**) Boy has taken other child’s toy [Action 2]; child who was pushed starts crying and girl asks her mother for help [Prosocial Action 4]. (**c**) Mother attempts to apply physical punishment to the child who had taken the toy [Action 3]. (**d**) Mother talks to child as strategy to resolve conflict [Prosocial action 4].

**Figure 2 ijerph-19-11561-f002:**
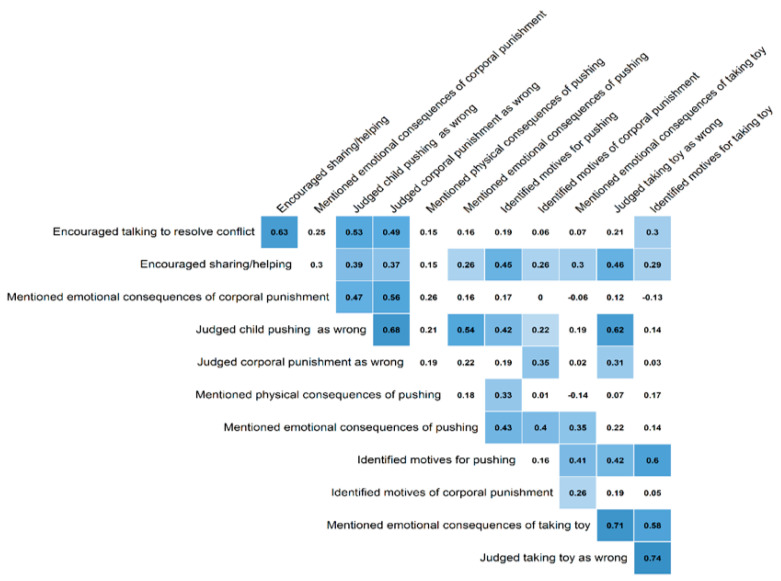
Tetrachoric correlation matrix of the 12 binary items used in the exploratory factor analysis. Note: Blue squares represent significant positive correlations. Darker colour tones represent larger correlation coefficients. White squares represent non-significant correlation coefficients at *p* < 0.05.

**Table 1 ijerph-19-11561-t001:** Coding system for maternal moral communication.

CODING ITEM	EXAMPLES OF COMMUNICATION
**Action 1: Pushing another child**	
**Judged the behaviour to be wrong**	*“That’s a wrong thing to do”; “You can’t do that”; “Fighting is nasty”*
**Suggested reasons why the child behaved like that**	*“He pushed the boy to get his bucket because his own had broken.”*
**Attributed punitive consequences to the behaviour**	*“When you do something wrong, you’re in trouble”* *“Mummy went to argue with him, see.”*
**Attributed physical and/or material consequences to the behaviour**	*“He got hurt because the other one hit him.”* *“The boy pushed him and went off with his little bucket.”*
**Attributed emotional consequences to the behaviour**	*“He’s sad because the other little boy hurt him”* *“See, the little boy was sad.” “Mummy’s going to be upset with him.”*
**Action 2: Taking another child’s toy without asking**	
**Judged the behaviour to be wrong**	*“You can’t take your little friend’s toy without asking; it’s nasty.” “He took the other child’s toy; see, that’s wrong.”*
**Suggested reasons why the child had behaved like this**	*“He wanted the little friend’s bucket because his broke.”*
**Attributed punitive consequences to the behaviour**	*“He took his little friend’s toy, and he’s going to be in trouble later because you can’t do this.” “Mummy went to argue with him because he took his friend’s toy, you see.”*
**Attributed physical and/or material consequences to the behaviour**	*“He was left without his little bucket, and now he can’t play anymore.”*
**Attributed emotional consequences to the behaviour**	“*He was sad without his little bucket, and he’s crying*”
**Action 3: Maternal aggressive discipline**	
**Judged the behaviour to be wrong**	*“Mummy can’t hit her son; it’s wrong”*
**Suggested reasons why the mother behaved like that**	*“Mummy was really so angry that she was going to hit him.” “She was going to hit her son because he did something wrong”*
**Attributed punitive consequences to the behaviour**	*“Mummy was going to hit her son, but the policeman was watching her.”*
**Attributed physical and/or material consequences to the behaviour**	*“If mummy hits her son, she’s going to hurt him.”*
**Attributed emotional consequences to the behaviour**	“*He’s sad because mummy wanted to hit him”*
**Actions 4: Encouragement of prosocial behaviour**	
**Encouraged conversation-based strategies to resolve conflicts**	*“You should apologise, shouldn’t you?” “You should talk.”*
**Encouraged sharing or helping behaviour**	*“You should share the toys.”* *“You should be friends and ask to borrow things.”*

**Table 2 ijerph-19-11561-t002:** Sociodemographic description of the sample of mothers. Pelotas, Brazil (*n* = 200).

	*n*	%
**Age (*n* = 200)**		
Under 20 years old	4	2.0
20 to 30 years	134	67.0
30 years or older	62	31.0
**Skin colour (*n* = 200) ***		
White	144	72.0
Black	32	16.0
Brown/mixed	21	10.5
Yellow	3	1.5
**Family income in minimum monthly wages (*n* = 199)**		
**Less than or equal to 1**	25	12.6
1.1 to 3	100	50.5
3.1 to 6	50	25.2
6.1 to 10	10	5.1
Greater than 10	13	6.6
**Schooling in full years (*n* = 172)**		
0 to 4 years	9	5.2
5 to 8 years	54	31.4
9 to 11 years	54	31.4
12 years or more	55	32.0
**Has a partner (*n* = 199)**		
No	49	24.6
Yes	150	75.4
**Worked outside the home since the child turned 2 years (*n* = 199)**		
No	65	32.7
Yes	134	67.3
**Self-perceived health (*n* = 195)**		
Excellent	26	13.3
Very good	30	15.4
Good	96	49.2
Fair	38	19.5
Poor	5	2.6
**Depressive symptoms (*n* = 199)**		
Normal	170	85.4
Moderate to severe	29	14.6

* Evaluated at the perinatal follow-up, otherwise child aged 4 years.

**Table 3 ijerph-19-11561-t003:** Frequencies of communication items and kappa concordance statistics between coders. Pelotas, Brazil (*n* = 200).

	*n*	%	Observed Agreement%	Kappa	SE **
**Action 1: Child pushing**					
**The mother judged the behaviour to be wrong.**			99.5	0.99	0.05
No	76	38.0			
Yes	124	62.0			
**The mother identified reasons for the behaviour.**			100.0	1.00	0.03
No	111	55.5			
Yes	89	45.5			
**The mother mentioned punitive consequences. ***			99.0	0.96	0.05
No	191	95.5			
Yes	9	4.5			
**The mother mentioned physical consequences.**			99.5	0.98	0.05
No	176	88.0			
Yes	24	12.0			
**The mother mentioned emotional consequences.**			99.5	0.99	0.05
No	72	36.0			
Yes	128	64.0			
**Action 2: Taking toy**					
**The mother judged the behaviour to be wrong.**			98.5	0.97	0.05
No	134	67.0			
Yes	66	33.0			
**The mother identified reasons for the behaviour.**			99.5	0.99	0.05
No	152	76.0			
Yes	48	24.0			
**The mother mentioned punitive consequences. ***			100.0	1.00	0.06
No	195	97.5			
Yes	5	2.5			
**The mother mentioned physical consequences. ***			100.0	1.00	0.06
No	192	96.0			
Yes	8	4.0			
**The mother mentioned emotional consequences.**			96.5	0.94	0.05
No	125	62.5			
Yes	75	37.5			
**Action 3: Corporal punishment**					
**The mother judged the behaviour to be wrong.**			96.5	0.94	0.05
No	140	70.0			
Yes	60	30.0			
**The mother identified reasons for the behaviour.**			99.5	0.99	0.04
No	134	67.0			
Yes	66	33.0			
**The mother mentioned punitive consequences. ***			98.5	0.96	0.06
No	197	98.5			
Yes	3	1.5			
**The mother mentioned physical consequences.**			98.0	0.95	0.07
No	200	100.0			
Yes	----	----			
**The mother mentioned emotional consequences.**			98.5	0.96	0.05
No	188	94.0			
Yes	12	6.0			
**Actions 4: Prosocial behaviour**					
**The mother encouraged talking to resolve conflict.**			99.0	0.98	0.05
No	164	82.0			
Yes	36	18.0			
**The mother encouraged sharing/helping.**			99.0	0.96	0.05
No	157	78.5			
Yes	43	21.5			

* Items removed in exploratory factor analysis. ** SE = standard error for the Kappa statistic.

**Table 4 ijerph-19-11561-t004:** Factor loadings of items concerning maternal moral communication (*n* = 200).

ITEMS	FACTOR 1 Interpersonal Moral Concerns	FACTOR 2 Material Moral Concerns
The mother judged child pushing to be wrong.	**0.845 ***	0.084
The mother identified reasons for pushing.	0.306 *	**0.495 ***
The mother mentioned physical consequences of pushing.	**0.288**	−0.011
The mother mentioned emotional consequences of pushing.	**0.432 ***	0.201
The mother judged taking toy to be wrong.	0.296 *	**0.770 ***
The mother identified reasons for taking toy.	−0.016	**0.850 ***
The mother mentioned emotional consequences of taking toy.	0.002	**0.769 ***
The mother judged corporal punishment to be wrong.	**0.815 ***	−0.205
The mother identified reasons for corporal punishment.	**0.309 ***	0.103
The mother mentioned emotional consequences of corporal punishment.	**0.652 ***	−0.294
The mother encouraged sharing/helping.	**0.635 ***	−0.001
The mother encouraged talking to resolve conflict.	**0.567 ***	0.219

** p* < 0.05. Note: Bold values indicate which factor items were assigned to.

## Data Availability

The procedure for accessing data from the Pelotas birth cohort studies, including those analysed in the current article, are detailed at http://www.epidemio-ufpel.org.br/site/content/studies/ (accessed on 1 June 2020).

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
