# Peer review of "Maternal Communication with Preschool Children about Morality: A Coding Scheme for a Book-Sharing Task"

_ijerph, 2022, doi:10.3390/ijerph191811561_

Round 1
Reviewer 1 Report
Very interesting and relevant scientific article.
The main question addressed in this research is development of a coding system for measuring mothers’ communication about morality with young children and testing its psychometric properties. Because authors used Book –Sharing task it seems that would be important to emphasize in the title of this research that authors work on coding shame based on Book-Sharing task (for example: Maternal Communication With Preschool Children About Morality: A Coding Scheme of Book-Sharing Task)
It seems that this topic is relevant in the field because there are small number of research about measuring mothers’ communication about morality with young children.
Compared to other publication, authors aimed to testing its psychometric properties of their coding system.
Regarding the methodology, it would be informative if the authors explain their statement: “The sample size did not allow for confirmatory factor analysis, 427
as well as exploratory factor analysis, to be conducted in this study. It is possible that some 428 items that we excluded from factor analyses because of their low frequency might be in- 429 included in larger samples.“ Because they made it anyway, therefore, an extra explanation is necessary for this analysis.
Conclusions are consistent with the evidence and arguments presented.
In some tables for example, there are typos. Also, for some tables for example “Table 3. Distribution of communication items and kappa concordance statistics between coders. Pelotas, Brazil (N = 200). “ Title is misleading, because there are no information about distributions.
Minor changes:
Please, write down references for those measures in sentence:
„However, less in known about the role of parental communications in conversations about moral issues, and few measures are available to assess parental moral communications to further this line of research.“
It seems that Ethical considerations is better to move before
It seems that Table 4 and Figure 2. contain redundant information.
Please, write how much is Inter-rater reliability for coding the 12 items!
Authors sometimes write subsample and sometimes sub-sample!
Please, maintain the letter size cohesive in the whole text-
I am not sure if paragraph Sociodemographic and Health Characteristics (170 ) is needed, because, from that paragraph we don not know anything about sample sociodemographic characteristic, we only learn what were questions for participants.
I am not sure if „stealing“ is appropriate world for situation where one child taking the other child's toy without asking.
Author Response
Response to Reviewer 1 Comments
We thank the reviewer for their comments. Please find our responses below.
Point 1: The main question addressed in this research is development of a coding system for measuring mothers’ communication about morality with young children and testing its psychometric properties. Because authors used Book –Sharing task it seems that would be important to emphasize in the title of this research that authors work on coding shame based on Book-Sharing task (for example: Maternal Communication With Preschool Children About Morality: A Coding Scheme of Book-Sharing Task)
Response 1: We agree and have changed the title to: Maternal Communication With Preschool Children About Morality: A Coding Scheme for a Book-Sharing Task.
Point 2: Regarding the methodology, it would be informative if the authors explain their statement: “The sample size did not allow for confirmatory factor analysis, 427 as well as exploratory factor analysis, to be conducted in this study. It is possible that some 428 items that we excluded from factor analyses because of their low frequency might be in- 429 included in larger samples.“ Because they made it anyway, therefore, an extra explanation is necessary for this analysis.
Response 2: We thank the reviewer for spotting this error. The sample size of course allowed exploratory factor analysis, as it was the focus of the present study. We felt, however, that running a confirmatory factor analysis with the same participants would be of limited value. Similarly, given the exclusion of some items due to low frequencies as well as low frequencies for some included items, we felt that it was not appropriate to run a confirmatory factor analysis with a subsample. We have now changed the sentence to: "We were not able to run additional confirmatory factor analyses, as it was not judged appropriate to use the same participants in both models. Furthermore, as some items already had low frequencies, we decided not to split the dataset into subsamples to confirm our results using confirmatory factor analysis." (page 15)
Point 3: In some tables for example, there are typos. Also, for some tables for example “Table 3. Distribution of communication items and kappa concordance statistics between coders. Pelotas, Brazil (N = 200). “ Title is misleading, because there are no information about distributions.
Response 3: We thank the reviewer for spotting this error. We changed the title of Table 3 to: “Table 3. Frequencies of communication items and kappa concordance statistics between coders. Pelotas, Brazil (N = 200).” (page 9).
Point 4: Minor changes: Please, write down references for those measures in sentence: However, less in known about the role of parental communications in conversations about moral issues, and few measures are available to assess parental moral communications to further this line of research.”
Response 4: We have added the following references:
Recchia HE, Wainryb C, Bourne S, Pasupathi M. The construction of moral agency in mother–child conversations about helping and hurting across childhood and adolescence. Developmental Psychology. 2014;50(1):34
Zych I, Gómez-Ortiz O, Fernández Touceda L, Nasaescu E, Llorent VJ. Parental moral disengagement induction as a predictor of bullying and cyberbullying: mediation by children’s moral disengagement, moral emotions, and validation of a questionnaire. Child Indicators Research. 2020;13(3):1065-83.
Point 5: It seems that Ethical considerations is better to move before
Response 5: We have moved the paragraph Ethical considerations to page 4, above the second paragraph of section Methods.
Point 6: It seems that Table 4 and Figure 2. contain redundant information.
Response 6: We are grateful for reporting this. We chose the keep only Table 4 and added the note: “Bold values indicate which factor items were assigned to” (page 11).
Point 7: Please, write how much is Inter-rater reliability for coding the 12 items!
Response 7: We added the following sentence: “Considering inter-rater reliability of each of the individual 12 items, the observed levels of agreement ranged between 96.5% and 100%, and the kappa statistic ranged from 0.94 to 1.0.” (page 11)
Point 8: Authors sometimes write subsample and sometimes sub-sample!
Response 8: We have amended the text and now only use “subsample”.
Point 9: Please, maintain the letter size cohesive in the whole text
Response 9: We reviewed the manuscript to maintain consistent font size.
Point 10: I am not sure if paragraph Sociodemographic and Health Characteristics (170 ) is needed, because, from that paragraph we do not know anything about sample sociodemographic characteristic, we only learn what were questions for participants.
Response 10: We agree with this point and have removed the relevant paragraph.
Point 11: I am not sure if „stealing“ is appropriate world for situation where one child taking the other child's toy without asking.
Response 11: We changed “stealing” to “taking away without asking” (page 1,5, 14, 15)

Reviewer 2 Report
For this topic, early childhood is a bit unknown population, unfortunately. That is the reason why this article is interesting.
Regarding the text, some reviews are necessary.
- Be careful with spaces between words (lines 65, 82, 85, 105, 119, 121, 265, 372).
- Maintain the letter size cohesive in the whole text (lines 139, 316-322).
- From my point of view, prosocial behaviour examples (action 4 Table 1) are incomplete. The usual way of communicating with children is as these examples show: “you have to talk; you have to apologise; you have to share, you have to be friends…”. All of them are based on orders and obligations.
I miss the research about more accurate methods of solving conflict based on:
a) the stimulation of empathy (how do you think the other boy is feeling?)
b) solution alternatives through the dialogue with the victim (what do you need to be repaired? What are you willing to do to repair the damage?).
This issue is the most relevant limitation and the need for future studies.
Author Response
Response to Reviewer 2 Comments
We thank the reviewer for their comments. Please find our responses below.
Point 1: Regarding the text, some reviews are necessary. Be careful with spaces between words (lines 65, 82, 85, 105, 119, 121, 265, 372). Maintain the letter size cohesive in the whole text (lines 139, 316-322).
Response 1: We thank the reviewer for spotting these errors. We have revised the manuscript accordingly.
Point 2: From my point of view, prosocial behaviour examples (action 4 Table 1) are incomplete. The usual way of communicating with children is as these examples show: “you have to talk; you have to apologise; you have to share, you have to be friends…”. All of them are based on orders and obligations.
Response 2:Thank you very much for bringing this to our attention. We re-examined these examples with a bilingual colleague and perceived that the emphasis on orders was the result of poor translation. We have adjusted the translation to reflect that mothers tended to communication about what “should be done” – moral encouragement, rather than giving orders.
We believe that these do capture the important moral communications sometimes expressed by mothers about prosocial behaviours. We judged that these “encouraging” communications were more relevant to our goal of assessing moral communication, than when mothers just described the prosocial behaviours as having occurred – without any recognition that they are a good thing to do, or that they should be done in these situations.
Point 3: I miss the research about more accurate methods of solving conflict based on: a) the stimulation of empathy (how do you think the other boy is feeling?) b) solution alternatives through the dialogue with the victim (what do you need to be repaired? What are you willing to do to repair the damage?). This issue is the most relevant limitation and the need for future studies.
Response 3: We agree with the importance of these issues. We emphasize that the objective of this article was focused on more specific aspects of moral communication (expressions of judgement of acts as wrong and why, and encouragement of prosocial behaviour), because less is known about this. Regarding narratives related to empathy and recognition of emotions, we understand that this type of narrative concerns the elements of the Theory of Mind, which does relate to moral development, even if the communication itself is not explicitly a moral judgement or communication of encouraging one type of social behavior over another. In parallel to the current study on specifically moral communications, there is separate evaluation of the narratives in these interactions in terms of mental state talk related to Theory of Mind. As the objective of this article was the development of the coding system with elements of moral communication, we chose to focus only on these elements and how it was developed, to contribute to this area.
